# Multi-Sensor Approach Combined with Pedological Investigations to Understand Site-Specific Variability of Soil Properties and Potentially Toxic Elements (PTEs) Content of an Industrial Contaminated Area

Simona Vingiani [1,2,*], Antonietta Agrillo [1], Roberto De Mascellis [3], Giuliano Langella [1,2], Piero Manna [3], Florindo Antonio Mileti [2] and Fabio Terribile [1,2]

[1] Department of Agricultural Sciences, University Federico II of Naples, 80055 Portici, NA, Italy; antonietta.agrillo@unina.it (A.A.); giuliano.langella@unina.it (G.L.); fabio.terribile@unina.it (F.T.)

[2] CRISP, Interdepartmental Research Centre on the "Earth Critical Zone" for Supporting the Landscape and Agroenvironment Management, University Federico II of Naples, 80055 Portici, NA, Italy; florindoantonio.mileti@unina.it

[3] Institute for Mediterranean Agricultural and Forestry Systems (ISAFOM), National Research Council (CNR), 80055 Portici, NA, Italy; roberto.demascellis@cnr.it (R.D.M.); piero.manna@cnr.it (P.M.)

[*] Correspondence: simona.vingiani@unina.it; Tel.: +39-081-253-9180



**Featured Application: Fast, cost efficient and non-invasive assessment of spatial variability of soil chemical and physical degradation phenomena.**

**Abstract:** A combination of indirect soil investigation by proximal soil sensors (PSS), based on geophysical (ARP, EMI), physical (Cone Index –CI– by ultrasound penetrometry) and spectrometric (γ-rays) techniques, as well as pedological surveys, was applied in the field to assess the spatial variability of soil pollution and physical degradation in an automobile-battery recycling plant in southern Italy. Five homogeneous zones (HZs) were identified by the PSS and characterized by soil profiles. CI measurements and field analysis showed clear features of physical (i.e., soil compaction, massive structure) degradation. XRF in situ (on profiles) analysis using portable equipment (pXRF) showed Pb, Cd and As concentrations exceeding the contamination thresholds provided by the Italian regulation for industrial land use up to 20 or 100 cm of depth. Hence, a validation procedure, based on pXRF field survey, was applied to the PSS approach used for the HZs identification. High consistency was found between the HZs and the PTEs in the most contaminated areas. Significant negative Pearson correlation coefficients were found between γ-rays dose rate and Pb, Cu, Zn, As and Ni; positive ones were found between γ-rays and autochthonous lithogenic elements (V, Ti, Mn, K, Sr, Nb, Zr, Rb, Th), confirming that higher radionuclide activity correlated with lower pollution levels.

**Keywords:** proximal sensors; soil pollution; PTE; pedological characterization; EMI survey; portable XRF

## 1. Introduction

The International Soil Reference and Information Centre (ISRIC) and the United Nations Environment Programme (UNEP), in their unique global estimate of soil pollution for the 1990's, approximated the extension of soil pollution at 22 million hectares [1]. In the last few decades, 80,000 sites in Australia were estimated to suffer from soil contamination [2], while in China, the Chinese Environmental Protection Ministry reported that 16% of all soil has been categorized as polluted; 19% percent was agricultural soil [3]. In 2013, the US EPA reported that 1300 contaminated sites appeared on the Superfund National Priorities List of the USA [4]. In the European Economic Area and the West Balkans, 2.8 million potentially polluted sites were estimated [5]. The point-source of soil pollution for two-thirds of those were due to industrial and commercial activities, such as waste disposal

and treatment [6]. However, the Global Assessment of Soil Pollution Report [7] the most recent update of global status in matters of soil contamination, included additional data for Eastern Europe, Africa, Latin America, the Caribbean and Asia; it was very limited, and then underestimated, in the previous census [8].

Hence, in this environmental scenario, there is an urgent need to achieve fast identification of spatial variability of environmental contaminants to apply targeted prevention and remediation strategies. Remote sensing techniques are alternative and efficient noncontacted detection methods for mapping and monitoring various soil and sediment contaminants [9,10]. They are essential tools, well-suited for surveying large areas, and monitoring soil contamination at a high temporal and spatial interval. As such, they can serve as a crucial tool in pollution detection [11,12] and ecological risk monitoring [13]. Despite the progress made on remote sensing techniques for soil studies, several crucial problems related to data acquisition—in particular environmental situations, i.e., the presence of vegetation on soil, clouds and shadows, etc.—have not been solved, and the interpretation of different surface variables from satellite data is compromised at temporal and spatial data scales [6,14–17].

Proximal soil sensors (PSS) are efficient and accurate techniques for measuring in-field variations of soil properties at a very fine spatial scale. They are more and more frequently required in environmental studies (i.e., soil pollution, land degradation processes, etc.) and precision farming (i.e., viticulture zoning). PSS are noninvasive, time- and cost-efficient, and offer many advantages over traditional techniques (e.g., no use of environmental polluting acids, little need for sample preparation, simplicity of use and easy portability, wide dynamic range of elemental quantification). PSS help to overcome the limitation of spatially scarce data to achieve fast identification of the spatial variability of contaminants and to apply targeted prevention and remediation strategies. Recently, the number of PSS techniques has increased and a variety of sensors measuring different soil properties (including electromagnetic induction (EMI), electrical resistivity (ER), ground-penetrating radar (GPR), $\gamma$-rays emissions, radiometric and fluorometric analyses, portable XRF spectrometer (pXRF), etc.) have become available on international markets. These sensors, generally coupled with GNNS receivers, acquire georeferenced parameters to process and map easily. They are considered effective, rapid and nondestructive techniques for the measurement of physical and chemical properties of soil and sediments. The reliability of geophysical proximal sensors (EMI and GPR) has been exploited to design maps of signal anomalies in agricultural areas [18,19] and contaminated sites [20]. Additionally, pXRF spectrometer was widely used to predict levels of potentially toxic elements (PTEs) in several environmental matrices [21–28]. The most commonly PTEs found in soil are cadmium (Cd), lead (Pb), zinc (Zn), copper (Cu), nickel (Ni), manganese (Mn) and arsenic (As). They are trace elements distributed worldwide [29–34], potentially causing hazards to the environment and human health at low concentrations. Maps of PTEs, based on LUCAS 2009 PTE data, are available for the European Union (EU28, except Croatia) [35].

However, in studies for purposes related to characterization plans and sustainable land reclamation strategies to be adopted in polluted sites, maps obtained by proximal sensors always need to be integrated by in situ pedological investigations to identify the soil properties related to specific signal or anomalies—as well as the variety of actions to be performed in polluted sites, such as identification, sampling and analysis of contaminated soil/materials, following official regulations. A recent regulation [36], which integrated the Italian Environmental Text [37] in matters of remediation strategies, environmental restoration and safety measures, provided geophysical surveys (electromagnetic induction (EMI) or electrical resistivity (ER)) for use in sampling in unhomogeneous areas or areas of unknown homogeneity (homogeneity is intended in terms of pedological characters and current or previous agronomic practices). The aim was to identify anomalous areas that would then be analyzed by direct excavation of pits or soil profiles.

Despite that, many questions have arisen about the most efficient method to be adopted for a preliminary assessment of spatial variability of unknown anthropogenic

soil contaminants or physically degraded soils, since the available sensors differ in their detected physical parameters. In a complex framework, such as that of contaminated sites, it is difficult to know in advance which method of prospecting will perform best. Combinations of soil sensors should be applied to identify those that best meet site-specific properties and issues, since in some circumstances, commonly-used EMI sensors can fail to distinguish between contrasting soils [38,39].

Hence, in an industrial site in southern Italy, formerly affected by anthropogenic waste disposal due to the activity of a Pb-battery recycling industrial plant, this work first aimed at tuning a multi-sensor methodology supported by pedological survey (i) to elucidate the spatial variability of peculiar soil properties related to soil degradation processes, including physical compaction and contamination by PTEs; (ii) to identify homogeneous zones (HZs) in terms of soil degradation, to be used for site-specific remediation practices and soil management; and (iii) to evaluate differences and connections between HZs and soil types/properties. Therefore, the approach provided by the Italian regulation [36] (i.e., EMI and ARP surveys associated with pedological investigations) was applied for the identification of HZs for soil pollution levels and physical degradation issues; however, additional PSS, including γ-rays spectrometry and ultrasonic penetrometry, were used. Then, a validation procedure, based on pXRF measurements of the main PTEs content at the field scale, was applied to test the effectiveness of the multi-sensor approach.

## 2. Materials and Methods

### 2.1. Geographical Setting

The study site was in southern Italy, in the town of Marcianise (province of Caserta), in a lowland environment of the Campania Plain (Figure 1). The proximity of this area to Phlegrean Fields and the Somma-Vesuvius complex, two of the most active volcanic centers of Italy, strongly affected pedogenetic processes and gave rise to the formation of Andosols and andic soil sequences. These alternated with volcanic deposits all over the Campania Plain [40,41], dating back some thousands of years [42]. Both Andosols and andic soils are known worldwide for their extreme fertility, as well as for their fragility with respect to pollution [43–45] and land degradation processes [46–48] in both lowland and mountain ecosystems [49] of Italy. The fertility of these pedo-environments makes them very attractive for agricultural land use, promoting intense anthropic spread with important environmental consequences on local communities.

In the study area, deep andic volcanic soils were frequently reworked on the surface by alluvial processes by rivers. Therefore, following IUSS Working Group WRB [50], soils around Marcianise were classified as Calcari-Vitric Cambisols [40].

The site was a 3.5 ha industrial area inside an automotive battery recycling plant operating since 1970. It was formerly utilized for temporary storage and improper disposal of Pb-batteries and related components, which progressively polluted the site with Pb, Sb, Cd and As. By the end of 2015, a soil pollution monitoring campaign was carried out by the University of Naples Federico II, aiming to assess the extent and spatial variability of contamination by PTEs. Some historic information on the site land use and their modifications over time (e.g., placing/removal of materials) were partially known due to recordings. Aerial photos obtained from the National Geoportal (WMS services) and Google Earth were collected for the area of interest, covering a period from 1989 to the present, to identify recent changes that could help identify site modifications. During the period of investigation considered in this article (October 2015–January 2016), some modifications (i.e., earthmoving, leveling) occurred in the site, and soil surveys were carried out over the whole period.

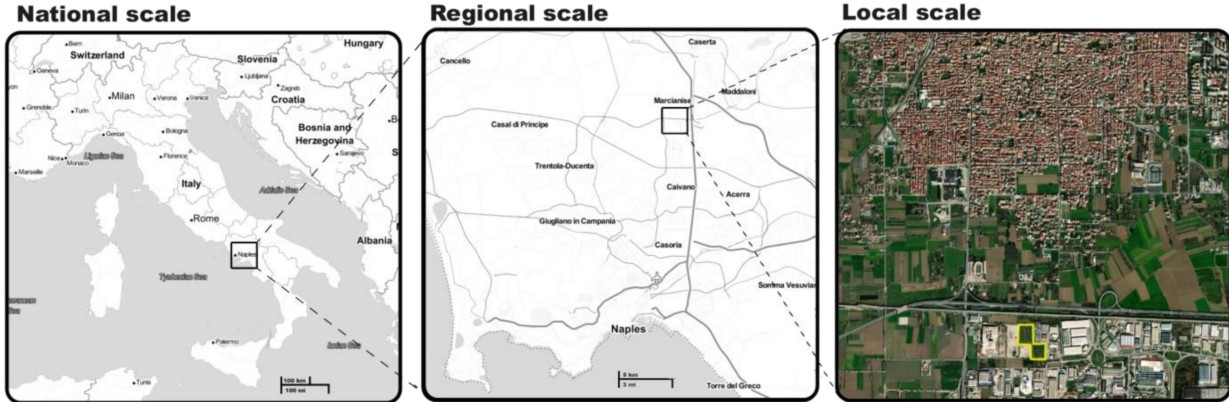

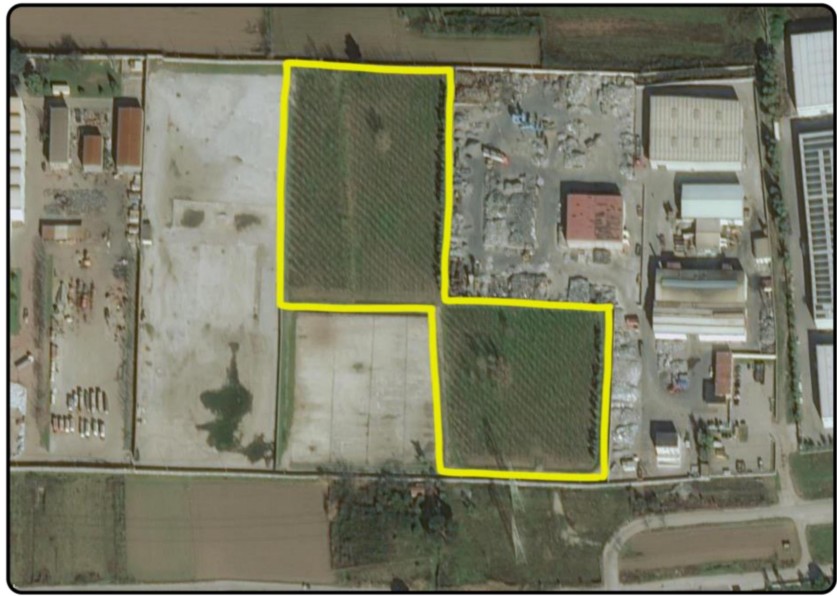

**Figure 1.** Sketch map of the study site at national, regional, local and field scales.

### 2.2. Geophysical Sensors

Automatic Resistivity Profiling (ARP) and frequency domain electromagnetic induction (EMI) represent the most potentially useful methods for preliminary soil characterization. ARP equipment (developed by Geocarta SA, Paris, France) allows the measurement of apparent electrical resistivity (ERa) of soils resulting from all soil resistances in the investigated volume. A mobile multi-electrode system, consisting of several electrodes mounted on 8 gear wheels, was towed by a quad and progressively made to advance on the soil surface with a signal acquisition speed of 8 m/s. Three different distances between dipoles (0.5, 1.0 and 1.7 m) allowed the simultaneous survey of three depths in a vertical section.

EMI methods in the frequency domain were used to measure the apparent electrical conductivity (ECa) of the subsurface (approximately from 1 to 10 m of depth). Two portable instruments (DUALEM 642-S and PROFILER EMP-400), transported by a sled pulled by a quad, acquired georeferenced (GSSI) data continuously along parallel trajectories. One day per technique was necessary for the survey. The use of DUALEM (Milton, ON, Canada) enabled measurements in two configurations (HCP and PRP), each characterized by three distances between coils, to obtain different investigated soil depths (1, 2, 3 m in PRP and 3.2, 6.4, 9.5 m in HCP). With the PROFILER, 3 frequencies (7 KHz, 10 KHz and 15 KHz) were measured in line VDM (vertical dipole mode) in a bandwidth extending from 1 kHz to 16 kHz in 1 kHz step, with a presumed investigated depth of approximately 1.8 m. Therefore, in the analyzed industrial site, ARP and DUALEM prospections were carried out before the main earthmoving and leveling works, when large heaps of waste were still

present in the site. Thus, the survey did not cover the entire soil surface. Furthermore, only the EMI survey was repeated after the leveling works, using a PROFILER EMP-400 (GSSI, Nashua, NH, USA).

### 2.3. Gamma Ray and Penetrometry Surveys

$\gamma$-rays and penetrometry prospections were carried out on a regular grid of points ($20 \times 20$ m$^2$) for 81 total points [22]. The GF Instruments Gamma Surveyor is a $\gamma$-rays spectrometer designed to measure natural and artificial radioisotopes in the ground, and in this work, was used to measure $\gamma$-rays dose rate, U, Th and K% to identify any anomalous increase (or decrease) in background radiation. Three minutes were used to set up for each measurement of $\gamma$-rays dose rate (81 points) and five minutes were used for the measurement of U, Th and K on a selection of points (19 points). The instrument used in this study was delivered with a factory calibration set to high-volume uranium standard. The total energy window was set from 0.12 to 3.00 MeV. Field measurements were carried out in static mode and the radioactivity was measured at discrete points. The detector was placed directly on the earth surface, to minimize the effects of local variation in relief and radioelement distribution. Spectrum and Assay modes were selected in the GF $\gamma$-rays spectrometer menu using a $^{137}$Cs $\gamma$-rays source, to stabilize the automatic gain control with an external GSSI ($\pm 4$ m). Measurement time was taken (180 s on each of the 81 points).

Then, an ultrasonic cone penetrometer (Rimik CP20) was used for the measurement and recording of cone index (CI) data on soils to the depth of 600 mm in field, with CI values up to 5 MPa. Penetrometers equipped with ultrasonic transducers calculated the depth of the probe and its sinking speed in real time, measuring the return time of a signal sent by the transducer, which bounced off a metal target placed near the ground and returned to the transducer. A data logger electronically recorded the force required to push the probe into the ground and the depth reading. For each point, 4–5 measurements in a radius of 0.5 m were acquired, to avoid erroneous interpretation due to fragments in soil. Collected data were informative of soil density and compaction.

Both $\gamma$-rays and CI data were interpolated and spatialized using the inverse distance weighting (IDW), 30 m radius, with QGIS 3.10, an open source software, developed by a team of dedicated volunteers, companies and organisations (www.qgis.org).

### 2.4. Pedological Survey

HZs were identified based on the ARP, EMI, $\gamma$-rays dose rate and ultrasound penetrometry surveys. A pedological characterization was carried out in each HZ by analyzing soil profiles dug to 160–200 cm depths, depending on the natural soil occurrence depth and the level of PTEs contamination. Soil horizons were described for morphological features (such as structure, color, roots and rock fragments content) following FAO [51] (2006), horizons and layers designated according with Soil Science Division Staff [52], and analyzed in situ for total element content by a pXRF analyzer (see details in Section 2.5). Then, they were sampled following pedogenetic/anthropogenic horizons [53]. Next, soil samples were air dried and sieved (<2 mm) for the main chemical and physical properties analyses (pH, organic matter, CEC, total carbonates, electrical conductivity) [54].

### 2.5. PXRF Measurements

A portable X-ray fluorescence analyzer (pXRF) was used to identify and measure PTEs at: (i) profile scale, in situ, at field conditions, on soil profiles, from the surface to a variable depth ranging from 70 and 185 cm; and (ii) on samples collected at 2 depths (0–15 and 15–40 cm) by hand drilling, following the regular grid designed for the systematic investigations carried out by $\gamma$-rays spectrometer and penetrometer surveys, with an additional 39 points (for a total of 120 points). Measurements on hand drilling samples were performed after drying and sieving (<2 mm) soils in lab. Scanning was performed with a Delta Professional (Olympus, DPO-4000) using an 8 mm$^2$ window. The instrument featured a Ta X-ray tube operating at 15–40 kV with integrated large area silicon drift

detector (165 eV). Innov-X software was used in Soil mode, the modality dedicated to trace element quantification, consisting of three beams operating sequentially, with acquisition times of 30 s per beam. Twenty elements (As, Ca, Cd, Cr, Cu, Fe, K, Mn, Nb, Ni, Pb, Rb, Sn, Sr, Th, Ti, U, V, Zn and Zr) were measured, on smooth uniform surfaces completely in contact with the instrument to minimize surface effects. Descriptive statistics of data measured with sampling on the regular grid, as well as data concerning quality checks on certified materials, are reported in Caporale et al. [22].

## 3. Results and Discussion

### 3.1. Spatial Distribution of Parameters Obtained by Geophysical Sensors

Data recorded by the ARP and the DUALEM surveys before the rearrangement works of the site (October 2015) were mapped, while zones occupied by heaps and trees corresponded to the parts of the maps that lacked measurements (Figure 2).

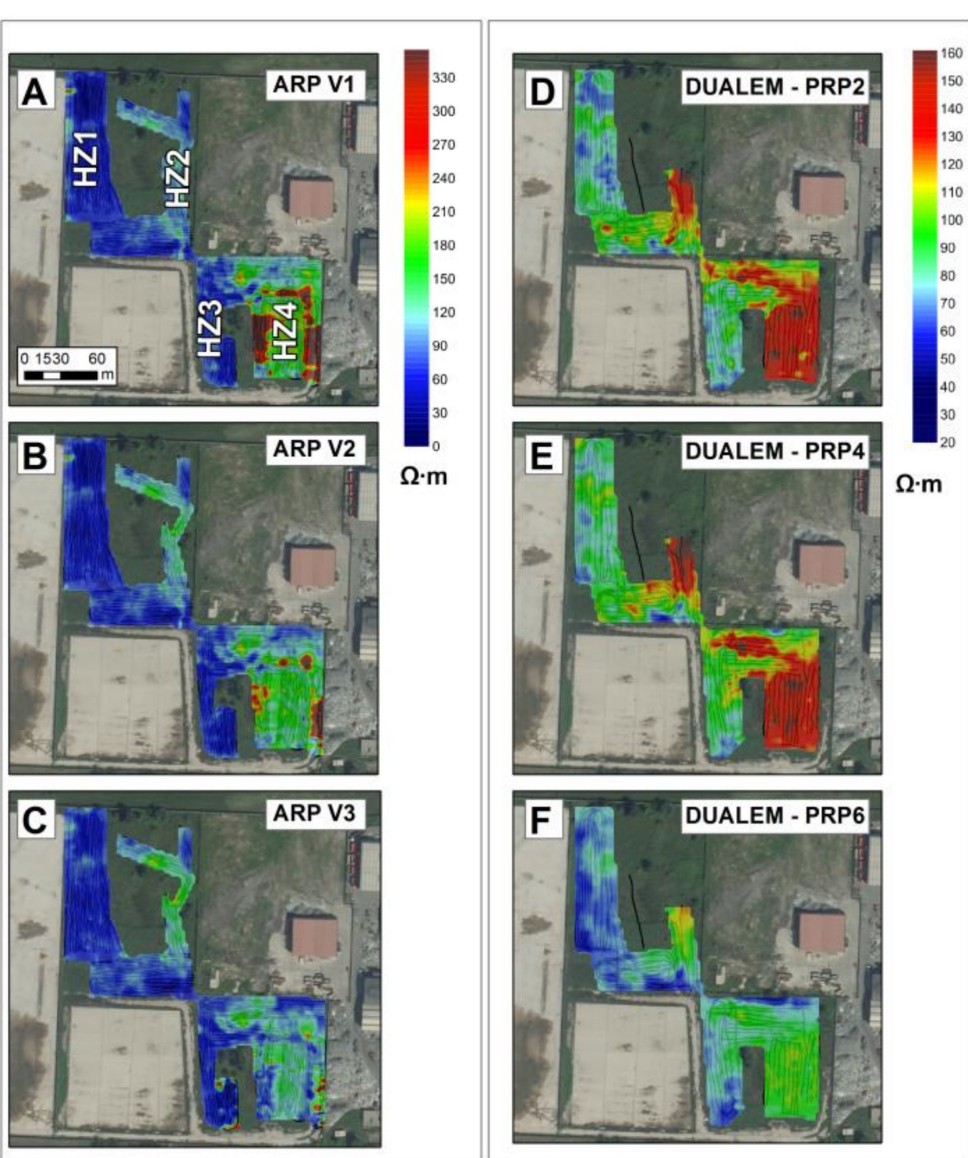

**Figure 2.** Maps of apparent electrical resistivity (ERa) obtained by ARP (**A**–**C**) and DUALEM (**D**–**F**) surveys for the investigation of different soil depths. (**A**) V1 = 0–0.5 m; (**B**) V2 = 0–1 m; (**C**) V3 = 0–1.7 m; (**D**) PRP2 = 0–1 m; (**E**) PRP4 = 0–2 m; (**F**) PRP6 = 0–3 m.

Descriptive statistics of data were reported in Table 1. In the ARP map corresponding to the surveyed depth from 0 to 0.5 m (Figure 2A), two homogeneous zones (HZs) were

identified in site 1, with HZ1 in the western part (an elongated N-S band with blue color) showing the lowest ERa values of the site (0–70 Ω·m), and the smaller HZ2 in the eastern part (an elongated N-S strip with light blue-green color) having values ranging from approximately 90–160 Ω·m. Moreover, two HZs were identified in site 2, including the HZ3 narrow band with low ERa in the western side (blue color) and the larger HZ4 in the mid-eastern part (green, yellow and red colors) with higher ERa values (varying from 180 and 340 Ω·m). These zones were clearly evident in the maps obtained by deeper surveys (0–1 m and 0–1.7 m), even though the highest ERa values of the wider HZ4 in site 2 decreased with depth (Figure 2B,C). By comparing ERa maps (Figure 2A,B) with archived pictures of the year 1998 (Figure 3A), the red narrow zone overlapped with an anthropic white colored form (Figure 3B), most likely a narrow street, which was no longer visible in the successive period, dated 2011 (Figure 3C). Looking at the DUALEM survey, the data showed a shorter range of value variability compared with ARP (0–180 and 0–330 Ω·m for DUALEM and ARP, respectively). Even though a high consistency was found between DUALEM and ARP maps, the DUALEM survey confirmed the HZs identified by the ARP (Figure 2).

**Table 1.** Descriptive statistics on ARP, PROFILER EMP and DUALEM data of geophysical surveys. ERa = apparent electrical resistivity; ECa = apparent electrical conductivity.

| | ARP | | | | PROFILER | |
| --- | --- | --- | --- | --- | --- | --- |
| | ERa VOIE1 | ERa VOIE2 | ERa VOIE3 | ECa 15 | ECa 10 | ECa 5 |
| | Ω·m | | | | KHz | |
| N. cases | 162,612 | 163,717 | 163,788 | 6441 | 6354 | 6146 |
| Mean | 113.5 | 98.4 | 88.2 | 17.7 | 17.5 | 17.1 |
| St. Dev. | 96.6 | 67.2 | 48.7 | 5.2 | 6.0 | 6.8 |
| Min | 1.1 | 1.2 | 2.4 | 1.1 | 0.0 | 0.1 |
| Max | 738.9 | 699.2 | 719.2 | 49.7 | 49.9 | 50.0 |
| Coef. Var. % | 85.1 | 68.3 | 55.2 | 29.2 | 34.4 | 39.4 |
| Skewness | 2.66 | 3.51 | 3.72 | 1.18 | 1.19 | 0.84 |
| Kurtosis | 8.37 | 6.25 | 4.12 | 4.83 | 4.87 | 3.21 |
| | DUALEM | | | | | |
| | ERa PRP2 | ERa PRP4 | ERa PRP6 | ERa HCP2 | ERa HCP4 | ERa HCP6 |
| | Ω·m | | | | | |
| N. cases | 36,688 | 36,278 | 35,780 | 35,285 | 36,504 | 36,618 |
| Mean | 94.8 | 105.0 | 77.9 | 90.7 | 79.9 | 70.7 |
| St. Dev. | 27.8 | 27.0 | 15.4 | 24.4 | 19.5 | 17.9 |
| Min | 20.2 | 46.9 | 36.4 | 34.1 | 27.5 | 23.1 |
| Max | 185.2 | 185.2 | 122.0 | 161.3 | 131.6 | 119.0 |
| Coef. Var. % | 29.3 | 25.7 | 19.8 | 26.9 | 24.4 | 25.3 |
| Skewness | 0.58 | 0.70 | 0.55 | 0.59 | −0.08 | −0.22 |
| Kurtosis | −0.39 | −0.12 | −0.07 | 0.00 | −0.03 | 0.07 |

However, in the DUALEM maps, no sign was found of the anthropic artifact identified with the ARP's survey in site 2.

Part of the rearrangement works for site 1 comprised the displacement of the materials stored as waste heaps in the middle of the site toward the western part, with the building of a long step approximately 40 cm in height, N-S oriented. Site 2 consisted in a random distribution of the waste materials on the topsoil.

A new geophysical survey was carried out with the PROFILER EMP and the measured ECa values at 15 and 7 KHz were spatialized (Figure 4), for an approximately investigated depth of 1.8 m (Figure 4A,B).

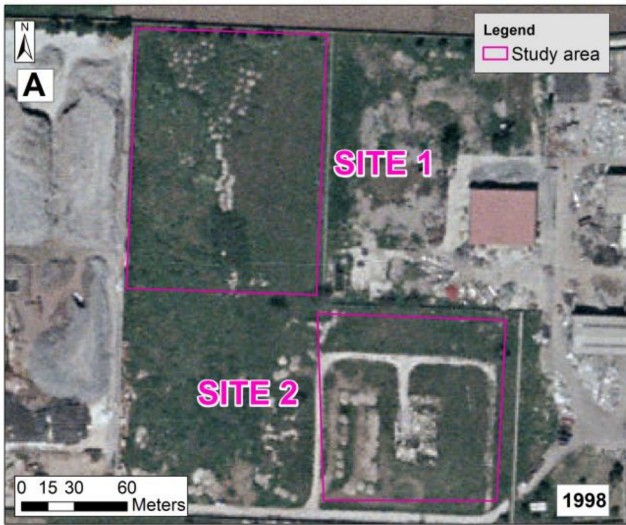

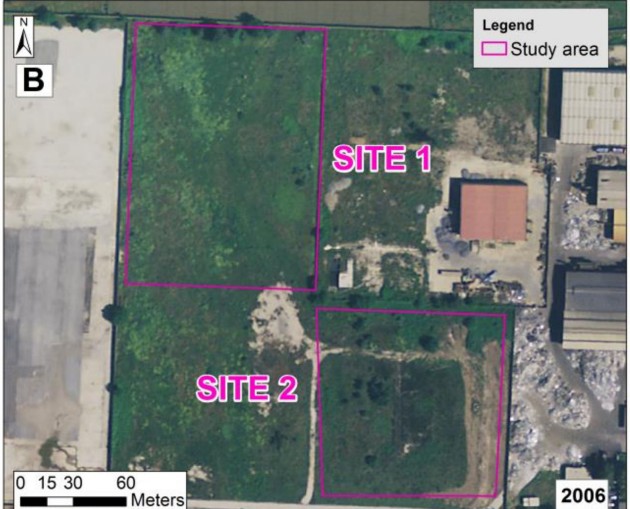

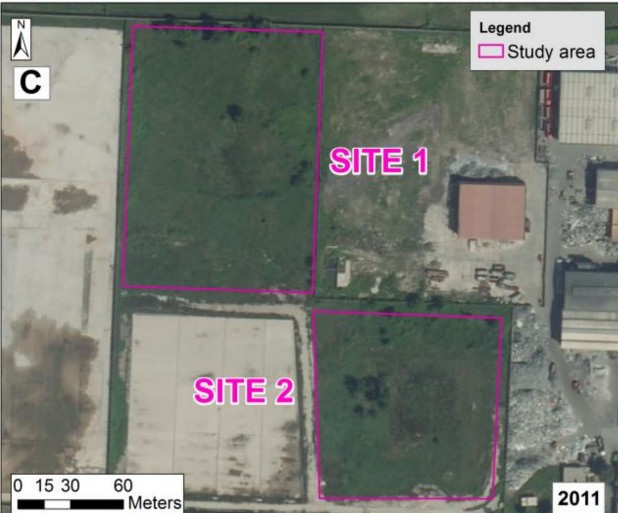

**Figure 3.** Archive pictures of the site in 1998 (**A**), 2006 (**B**) and 2011 (**C**).

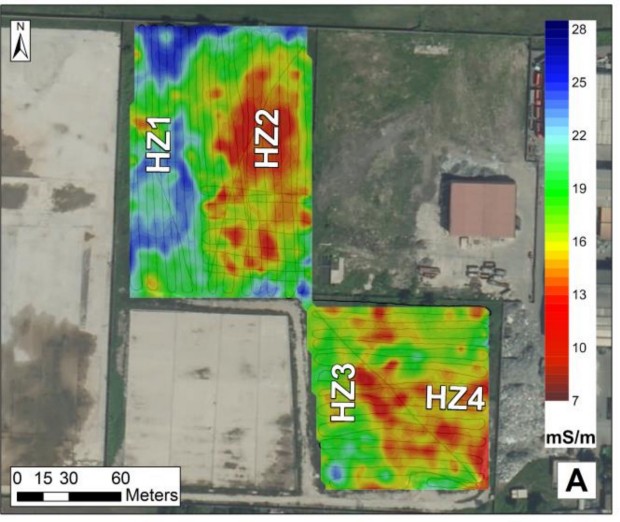

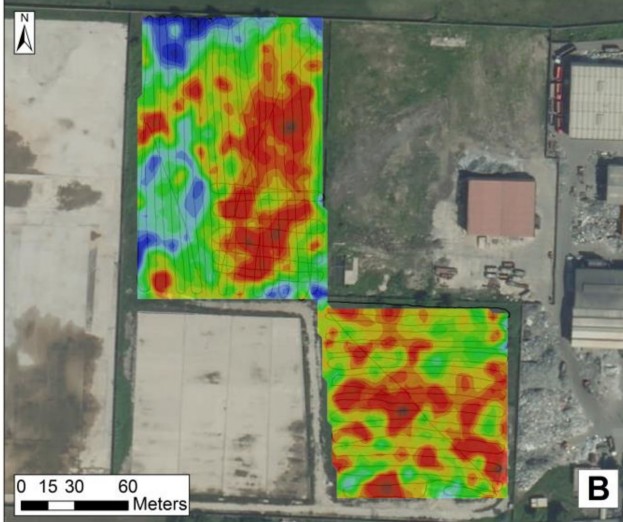

**Figure 4.** ECa maps (in mS/m) obtained from the PROFILER survey at (**A**) 15 KHz and (**B**) 7 KHz.

In both maps, two main HZs were identified in site 1: one in the western part (blue color), along the step and overlapping the previously identified HZ1, showing the highest ECa values of the site (19–28 mS/m), and another in the eastern part (red color), characterized by low ECa values (15–9 mS/m), formerly occupied by waste heaps and comprising HZ2. In site 2, a jeopardized pattern of ECa values added great difficulty to the identification of HZs, although a small HZ (i.e., HZ3) with N-S strip shape and middle values (16–19 mS/m) was recognized in the western part (green color, Figure 4A), while middle-low values (16–10 mS/m) were found in the mid-eastern part of site 2 (red color).

Therefore, through comparison of all the geophysical maps, high consistency arose for the two HZs of site 1 (HZ1 and HZ2) and the two HZs of site 2 (HZ3 and HZ4). However, the HZs were better defined in the maps of the ARP and DUALEM surveys (Figure 2), which were carried out in the phase preceding the rearrangement works of the site, compared with the PROFILER EMP maps (Figure 4) obtained by the survey carried out after the waste displacement. The effect of the site works on the geophysical signal showed by the PROFILER EMP maps was more evident in site 2, where the random distribution of the wastes on the soil surface very likely produced new geophysical anomalies.

Therefore, the results of the geophysical surveys by measurement of soil ERa and ECa enabled the identification of field heterogeneity due to deep in situ soil reworking and/or presence of polluted materials. Indeed, ERa and ECa are key parameters for understanding high spatial resolution soil geography, since they are strongly affected by physical, chemical and hydrologic properties, including soil moisture [55], soil compaction, coarse fragments and clay content [56,57], salinity [58] and carbonate content [59]. ECa maps represent the combined results of the spatial variability of the soil properties, but the effect produced by a single property is impossible to identify. Nevertheless, EMI sensors are widely applied in anthropogenic [20] and natural environments [25,26,59] due to their effectiveness in studies of soil spatial variability and identification of HZs for purposes related to pollution remediation and precision agriculture. As a general rule, PSS surveys are carried out only after earth movements and site rearrangements, enabling researchers to collect information on the state of such places. The EMI surveys conducted pre- and post-works gave us the opportunity to (i) better identify the HZs based on both preexisting and current soil physical conditions still affecting the present soil properties, and (ii) to monitor changes in the patterns of anomalies after earth movements in the field.

### 3.2. γ-rays Spectrometry and Penetrometry Surveys

By preliminary random soil surveys using of a hand auger, valuable signs of soil physical degradation (such as massive soil structures, the presence of dense/hard layers and coarse fragments starting from the topsoil) were identified throughout the site. Therefore, two additional PSS, including γ-rays spectrometry and ultrasound penetrometry, were applied, due to their high sensitivity to mineralogical and physical soil property variation. The data were used in combination with the ARP and EMI to define the HZs. Descriptive statistics of data are reported in Table 2. Potassium content ranged from 2.0% to 3.6%, U from 5.6 to 10.4 mg/kg and Th from 13.1 to 23.0 mg/kg. The γ-rays dose rate ranged from 60.4 to 121.9 nGy/h (Table 2). Pearson correlation coefficient underlined two significant tail correlations between γ-rays dose rate and the single elements (γ-rays dose rate vs. K = 0.798, vs. U = 0.685, vs. Th = 0.849), indicating that all these elements contributed to the radionuclide emission activity of the studied soils. Compared with measurements obtained from Vesuvius soils 20 km away as the crow flies (unpublished data), the studied area showed much lower values of K, U, Th and dose rate (in Vesuvius soils, K ranged from 3.9% to 5.1%, U from 24.1 to 31.66 mg/kg, Th 12.8 to 20.3 mg/kg and γ-rays dose rate from 233.5 to 279.0 nGy/h), very likely due to the dilution effect created by mixing with allochthonous wastes. Indeed, the natural origin of K, U and Th was reported in the literature in the volcanic soils of the Phlegrean Fields, Mt. Roccamonfina volcano area and Nola-Pomigliano area (north of Mt. Somma-Vesuvius volcano) [60,61]. The common origin was confirmed by the significant correlation between K and Th (0.757) in the studied site.

**Table 2.** Descriptive statistics of γ-rays dose rate and penetrometry survey (Cone index (CI)) data at different depths (0–10, 10–30, 30–45 cm).

| | K | U | Th | γ-Rays Dose Rate | CI | | | |
| | | | | | 0-10 cm | 10-30 cm | 30-45 cm | 45-60 cm |
| | % | mg/kg | mg/kg | nGy/h | Mpa | | | |
|---|---|---|---|---|---|---|---|---|
| N. cases | 19 | 19 | 19 | 81 | 81 | 81 | 81 | 81 |
| Mean | 2.9 | 8.2 | 17.9 | 97.6 | 2.21 | 2.90 | 3.60 | 3.93 |
| St. Dev. | 0.4 | 1.2 | 2.2 | 11.2 | 0.60 | 0.69 | 1.00 | 1.13 |
| Min | 2.0 | 5.6 | 13.1 | 60.4 | 0.77 | 1.63 | 1.90 | 1.58 |
| Max | 3.6 | 10.4 | 23.0 | 121.9 | 3.46 | 5.00 | 5.00 | 5.00 |
| Coef. Var. % | 14 | 14 | 12 | 11.5 | 27 | 24 | 28 | 29 |
| Skewness | −0.5 | −0.3 | 0.2 | −0.4 | 0.0 | 0.6 | 0.3 | −0.3 |
| Kurtosis | 0.1 | 0.1 | 1.2 | 0.9 | −0.6 | 1.2 | −1.3 | −1.6 |
| Pearson correlation coefficients | | | | | | | | |
| U | 0.2 | | | | | | | |
| Th | 0.757 ** | 0.284 | - | | | | | |
| γ-rays dose rate | 0.798 ** | 0.685 ** | 0.849 ** | - | −0.145 | −0.078 | 0.103 | −0.159 |

** Correlation significant at level 0.01 (two tails).

The georeferenced grid of the measurement points is shown in Figure 5A. Based on the data probability distribution, five classes of values were used to map the γ-rays dose rate variability (Figure 5B). Two main HZs were identified in site 1 (yellow and light green areas) with intermediate–low (78–100 nGy/h) and intermediate–high (100–117 nGy/h) values, respectively. Three HZs were identified for site 2 (red-orange, mainly yellow, and green areas), characterized by the lowest (60–90 nGy/h), intermediate (78–105 nGy/h) and the highest (117–122 nGy/h) values of γ-rays dose rate. By comparison of the γ-rays dose rate with the ARP and EMI maps (Figures 2 and 4), consistency was found in site 1 between the zone of intermediate–low γ-rays (Figure 5B) and HZ1, as well as between the intermediate–high γ-rays values and HZ2. For site 2, the lowest values of γ-rays dose rate were measured in HZ3, identified by the ARP and EMI surveys (Figures 2 and 4), while two subzones were defined for HZ4 (i.e., HZ4a and HZ4b). These were marked by intermediate and high γ-rays dose rates. As reported in the literature, higher levels of radionuclide activity concentrations are associated with igneous rocks, such as granite, and lower levels are associated with sedimentary rocks [62]. In particular, Kalyoncuoglu [63] demonstrated that the highest absorbed γ-rays dose rates for the Isparta plain (526.28 nGy/h) were in connection with Gölcük volcanics, while the lowest (18.70 nGy/h and 22.03 nGy/h) were found in the Davraz and Söbü limestones. Therefore, in our study, the areas where mainly autochthonous volcanic soils occurred, due to minimal soil/waste mixing processes, were expected to have higher γ-rays dose rate values. Additionally, thicker carbonatic layers and higher coarse carbonatic fragments content in the soil (or the presence of allochthonous nonvolcanic materials) were expected to correlate with lower γ-rays dose rates, despite the level of soil contamination.

Penetrometry ultrasound measurements were performed in soil depths ranging from 0 to 60 cm and results obtained in the ranges of 0–10 cm, 10–30 cm, 30–45 and 45–60 cm of depth were mapped separately. For the sake of brevity, only 0–10 and 45–60 cm maps have been reported herein (Figure 5C,D). The map of the most surficial layer (0–10 cm of depth) showed generally high cone index (CI) values (average CI value of 2.2 MPa), with 61% of measurements exceeding 2 MPa. These were mainly concentrated in site 1 (Figure 5C). Indeed, as shown by the map in Figure 5C, HZ1 (from A2 to A8 and from B2 to B8 points) had most of the highest CI values (5 MPa) of the layer, while HZ4b (site 2) had the lowest values, ranging from 1 to 2 MPa. Regarding higher depths, in the range of 10–30 cm, CI values increased very rapidly (mean CI value of 2.9 MPa), with 91% of the measurements exceeding 2 MPa. Meanwhile, at 30–45 cm, 99% of CI data exceeded 2 MPa and, among them, 28% of points reached the maximum CI value of 5 MPa. The deepest layer (45–60 cm)

showed a further increase of the CI mean values (3.9 MPa), although in 30% of points, a decrease of CI values was also detected. However, consistent with that observed for the most surficial layer, HZ4b of the 45–60 cm layer showed lower values of CI compared with the shallower areas.

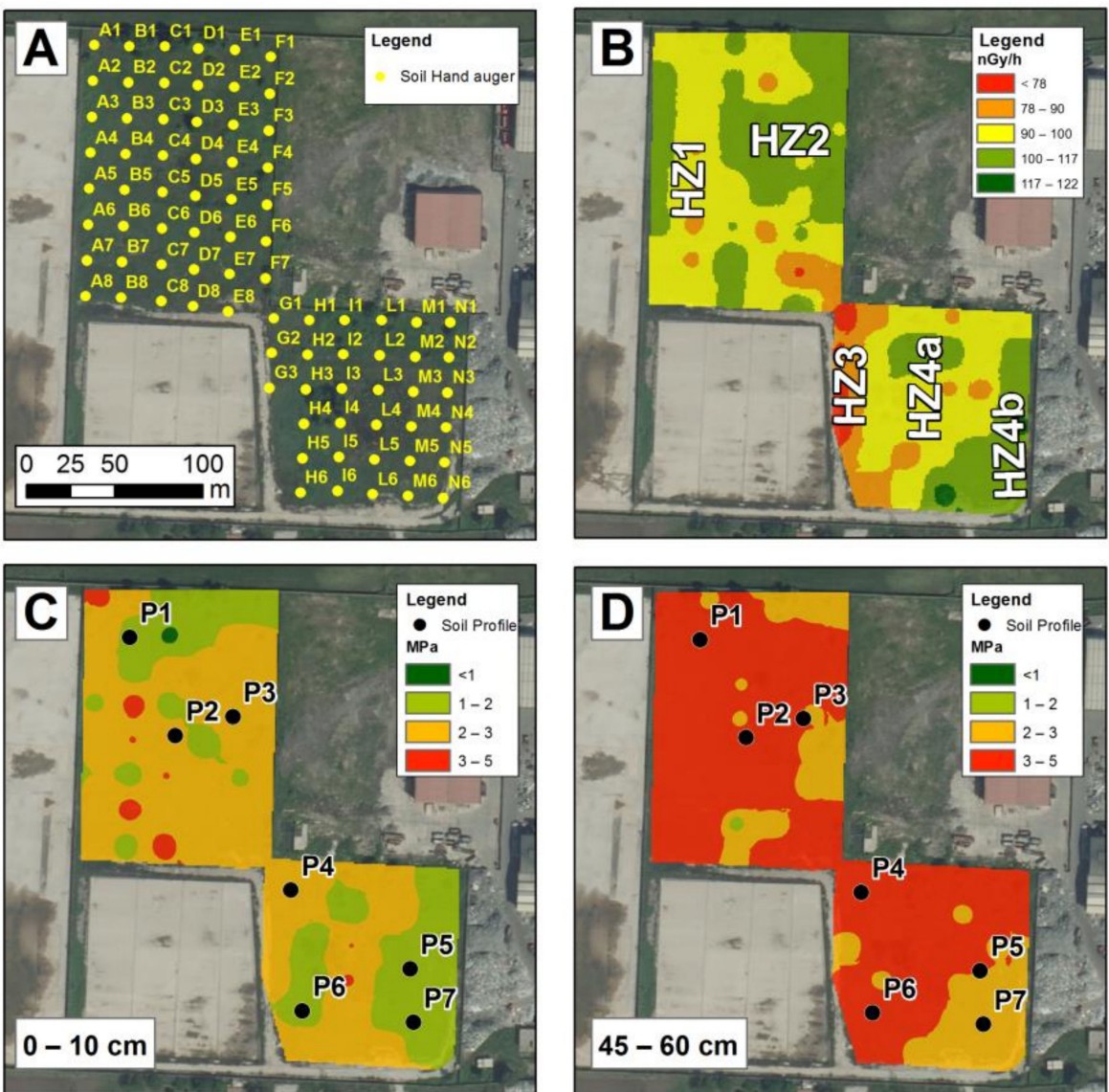

**Figure 5.** γ-rays dose rate and penetrometry ultrasound maps. (**A**) Sampling scheme; (**B**) γ-rays dose rate map; (**C,D**) cone index (CI) maps at 0–10 cm and 45–60 cm.

It is well known that the shear strength, penetration resistance and hydraulic properties of soils change according to the level of compaction. Both these mechanical and hydraulic properties influence the rate at which plant roots can grow to depth in a soil, and the flow and availability of water and nutrients for plant use. For a given CI, the amount of root growth reduction will vary among plant species and varieties as well as soil types [64]. However, CI classes were selected according to values reported in the literature for root reducing (1 MPa) and restricting (2 MPa) limits [65,66], defined by study of the percentage of cotton tap roots penetrating 2.5 cm thick layers of soil compacted to different degrees of cone penetration resistance pressures (PR) in four soil types. The range of 1–2 MPa was identified as a state of reducing penetration, 2–3 MPa as a state of restricting penetration, and the 3–5 MPa range was considered a hardened soil, very difficult to penetrate, even for tree roots. Therefore, by the evaluation of the CI maps at different soil depths, a picture of

high compaction and degradation of soil physical properties was outlined for the whole studied area, starting from the topsoil. Considering that the typical natural soils of these environments were volcanic ash soils, which have humus rich surface soils and light, fluffy, easy to break clods [67], the observed soil compaction had to be ascribed to anthropic causes, very likely to the downward forces of moving vehicles on the soil surface and on the trails of the worksite, as identified in archive pictures (Figure 3).

The Pearson bivariate correlation coefficient, calculated between the γ-rays dose rate and CI data (Table 2), did not demonstrate significant correlation. The lack of correlation suggested that soil compaction did not produce variability on γ-rays emissions. Indeed, natural emissions of γ-rays were mainly related to the parent material mineralogy [68,69]; thus, for soil with similar mineralogy, differences could be identified in soil texture variations [70] due to the high adsorption capacity of clay fractions for chemical elements and then radionuclides [71]. However, the correspondence on the map found between the lowest CI values at 0–10 cm and the highest γ-rays dose rate values of site 1 in HZ4b was of note—as was the presence of the highest CI values in HZ1, which had already been identified for its peculiar properties based on EMI and γ-rays dose rate measurements.

*3.3. Pedological and Chemical Properties of the Homogeneous Zones (HZs)*

A pedology-based approach was applied for the characterization of the HZs identified by the geophysical, spectrometric (γ-rays dose rate) and ultrasound penetrometry surveys. Five representative soil profiles (P1, P3, P4, P5 and P7) (Figure 6) were used to focus on the pedological properties of the 5 identified HZs: for site 1, P1 in HZ1 and P3 in HZ2; for site 2, P4 in HZ3, P5 in HZ4a and P7 in HZ4b. Two additional profiles were also performed: P2 on the boundary between HZ1 and HZ2, and P6 in the southern part of HZ3. Soil morphological and chemical properties are reported in Table 3A,B. Field morphological descriptions of the profiles showed massive hard structures in the surface horizons of all the soil profiles. In particular, P1, P2, P4 and P5 showed massive structure to a depth of 80–110 cm, while P3 and P6 were massive until 25–35 cm and P7 was not massive at all. Moreover, the profiles had generally common (5–15%) coarse fragments on the surface that decreased with depth, except for P1, which had abundant (21–26%) coarse fragments in the first 105 cm, formed by landfill materials (HZ1), and P7, which showed only volcanic clasts (pumices and scoria) below 65 cm. Few or no roots were detected in the soil profiles, except for P4 and P7, in which common roots were found in the topsoil and few roots were found up to 180 and 100 cm, respectively. Then, different soil colors characterized the soil profiles: (i) in site 1, P1 was the only profile showing mainly olive (5Y) colors, while both P2 and P3 were olive brown (2.5Y), except for the deepest horizon (175–185+) of P2, which turned to yellowish brown (10YR); (ii) in site 2, P4, P5 and P6 were olive brown until 70 or 95 cm and yellowish brown in the deeper buried horizons, while P7 was yellowish brown starting from the topsoil. It is known that ancient Andosols and andic soils of Campania volcanic landscapes typically show sequences of buried yellowish brown soils, due to the presence of organo-mineral complexes, poorly ordered iron oxides (including ferrihydrite) and short-range ordered clay minerals [40,41]. PTE contamination (as later shown in Table 4) was found only for olive and olive brown colored soil horizons, and never for yellowish brown horizons even when they formed the topsoil. Therefore, the soil color seemed to represent a good covariate of other soil properties (such as soil structure or clasts abundance) that could be used for fast surveys aiming to discriminate between allochthonous anthropogenic (polluted) and autochthonous volcanic sources. Anthropic discontinuous ˆCu layers at low depths (50–60 cm), made by abundant (approximately 70%) cement clasts for a thickness of approximately 20–25 cm or thin (6–7 cm) Cm massive calcareous layers, were found in P4 and P5, respectively; in both cases, they covered hardened compacted horizons. The caret symbol (ˆ) is a prefix that indicates horizons and layers formed in human-transported material. Indeed, presumably this material has been moved horizontally onto a pedon from a source area outside of that pedon by purposeful human activity, usually with the aid of machinery or hand tools. The "u" suffix indicates the

presence of objects or materials that have been created or modified by humans, typically for a practical purpose in habitation, manufacturing, excavation, or construction activities [52]. Therefore, these C layers were supposed to be the remnant of ancient, buried worksite trails for moving vehicles in the industrial site, which caused strong compaction of the underlying soil structure. Close to both P4 and P5, buried trails were identified in the archive picture from 1998 (Figure 3B), and these were already identified in the ARP maps (Figure 2A,B) as red zones. Therefore, considering that a decrease of radionuclide activity is generally measured in correspondence of limestone, carbonatic layers and stoniness, the presence of C layers, like those observed in P4 of HZ3 and P5 of HZ4a, could be related to the low values (71–77 nGy/h) of $\gamma$-rays dose rates in HZ3 (the lowest of the whole site) as well as those measured in proximity to P5 (86 nGy/h) in HZ4a. This would be in addition to the effects of the waste disposal. On the contrary, in area P7, where no stones were identified but evidence of volcanic soil was found, higher $\gamma$-rays dose rate values (110 nGy/h) were measured (Figure 5B). These results were consistent with data reported from Tuscany for soils formed by different parent materials and characterized by different stoniness levels. Soil reaction was generally neutral to slightly alkaline (from 6.7 to 8.2), while higher values (pH = 8.3–8.4) were measured for the massive topsoil and C horizon of P4. However, P1 and P4 showed the highest average pH values (7.9), as well as the highest total carbonates content (4.5–8.5%). P1 showed the highest electrical conductivity values. Soil organic C content had a general decreasing trend with depth, but when buried A horizons (Ab) were encountered, increasing content was measured. Correlation between CEC and OC was not strongly significant (0.419 *) because of the highly anthropized context. On the contrary, when the regression was calculated between CEC and OC, taking into account only the buried (Ab and Bwb) yellow-red (YR) and andic-like soils, a very high coefficient of determination was found ($r^2 = 0.88$, respectively), suggesting the major role of organic matter in soil charge for these natural soils. This result was consistent with the highest values of CEC (from 31.2 to 53.0 cmol (+)/kg) measured for these buried soils. It is known that Andosols and andic soils frequently show high values of CEC when measured with the $BaCl_2$ (pH 8.2) and triethanolamine methods, due to their peculiar highly variable and pH-dependent cation exchange capacity (CEC increasing with pH). This is caused by the presence of variable charge minerals, such as allophane and imogolite [72], and organic matter. In situ measurements of PTEs are reported in Table 4. All the topsoils (from the surface to 20–35 cm) except P7 were widely contaminated by Pb, from 1 to 10 times the CSC3; however, for P1, contamination reached 100 cm of depth (Table 4A). Cadmium was found to exceed CSC3 for the same horizons contaminated by Pb. Then, among the possible interelement interferences, the Olympus factory indicated that high levels of Pb could interfere with low levels of As. This element content, measured by XRF, was recalculated considering aqua regia (AR)—ICP-MS measurements [22] below 200 mg/kg, the maximum value reached by As in the profiles (Table 4A,B). Per the results, surface horizons were generally the most contaminated by As: in P3 and P4, As content exceeded CSC2 in depths ranging from the surface to 25 cm, while in all the other profiles, As exceeded CSC3 in the same range, but P1 showed As contamination exceeding CSC1 up to 150 cm (Table 4A,B). For Cu and Zn, several topsoils were found to exceed CSC1 and CSC2, while no CSC exceeding value was found for Cr. P7 was the only profile that did not exceed CSC3 for Pb content (Table 4B). It also showed the lowest As and Cd content (below CSC1) found among all the analyzed soil profiles. Significant correlations between PTE and chemical-physical soil properties were found between OC and Cu (0.475 **), Pb (0.446 **) and Zn (0.480 **), and between electrical conductivity (EC) and As (0.534 **), Cu (0.479 **), Pb (0.559 **) and Zn (0.556 **). No correlation was found between PTE and CEC. These results were consistent with findings that the most contaminated were generally the uppermost OC-enriched horizons, while the PTE correlation with EC was likely attributable to higher contents of soluble salts in these anthropogenic materials, primarily made by different types of batteries.

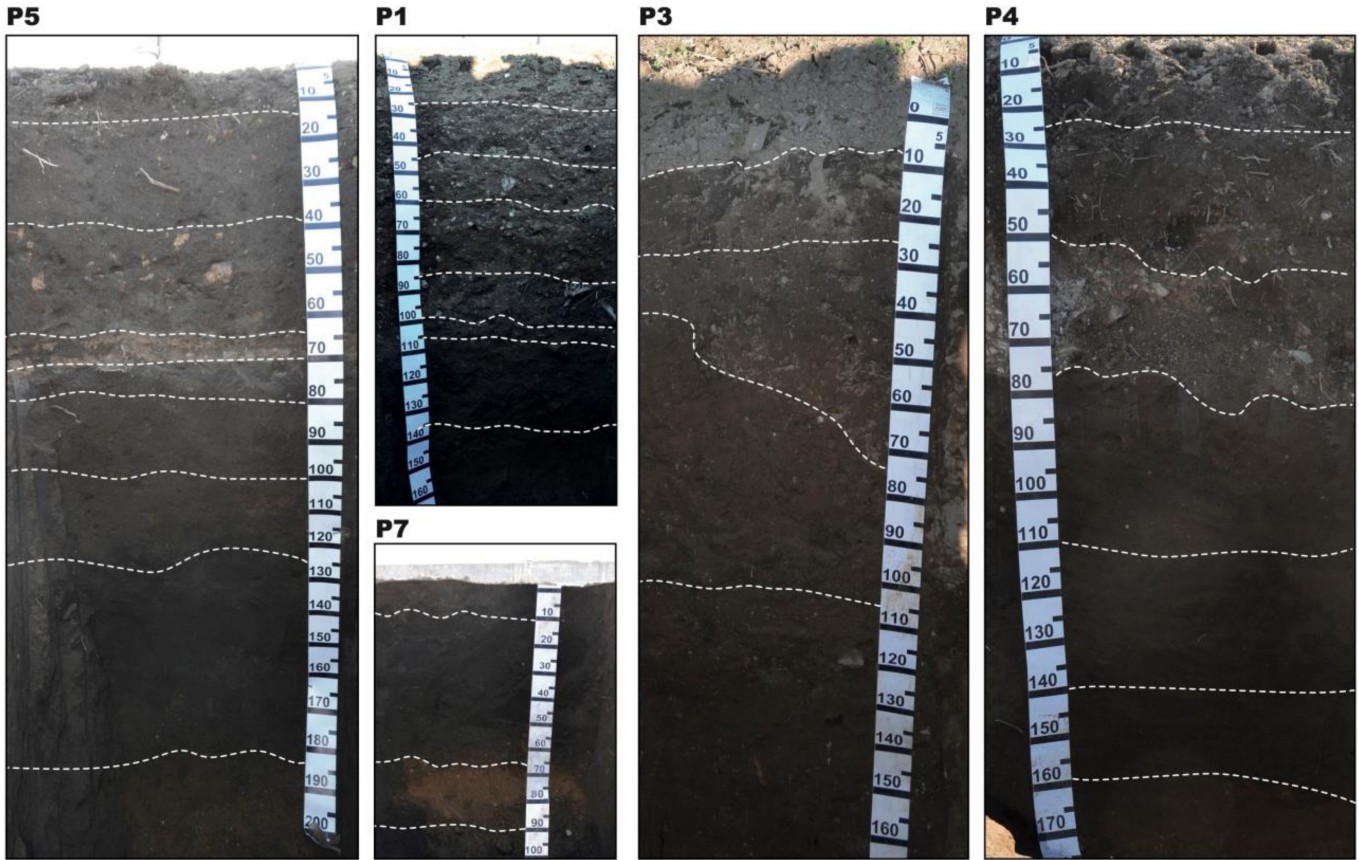

**Figure 6.** Soil profiles characterizing HZ1 and HZ2 in site 1 (**P1** and **P3**, respectively), and HZ3, HZ4a and HZ4b in site 2 (**P4**, **P5**, and **P7**).

**Table 3.** (**A,B**) Main chemical and morphological properties of soil profiles (P). ND = not determined. The caret symbol (ˆ) is used as a prefix to indicate horizons and layers formed in human-transported material.

| | | | | | | | | | | | | | | |
|---|---|---|---|---|---|---|---|---|---|---|---|---|---|---|
| | | | | | | | | | | | | | | |
| | | | | | | | | | | **(A)** | | | | |
| Profile | Soil Horizon | Depth (cm) Up | Down | Soil Color | Structure | Coarse Fragments % | Roots | OC g/kg | pH H₂O | KCl | μS cm⁻¹ | CEC cmol(+)/kg | TC % | EC % |
| | ˆAu | 0 | 25 | 5Y 5/3 | SG | 22 | VF, F | 14.9 | 7.8 | 7.0 | 461 | 20.2 | 6.7 | 6.7 |
| | ˆAum1 | 25 | 45 | 5Y 5/3 | MA | 24 | VF, F | 19.3 | 7.7 | 7.0 | 470 | 20.9 | 9.5 | 9.5 |
| | ˆAum2 | 45 | 60 | 5Y 5/4 | MA | 21 | VF, F | 11.7 | 8.2 | 7.0 | 223 | 25.0 | 4.8 | 4.8 |
| P1 | ˆAum3 | 60 | 85 | 5Y 4/4 | MA | 23 | F, F | 12.4 | 8.1 | 7.0 | 244 | 21.8 | 4.4 | 4.4 |
| | A | 85 | 100 | 5Y 4/2 | MA | 26 | F, F | 46.1 | 7.6 | 6.8 | 388 | 31.1 | 3.5 | 3.5 |
| | Ac | 100 | 105 | 5GY 4/2 | MA | 14 | F, F | 18.6 | 8.1 | 7.0 | 226 | 23.7 | 4.7 | 4.7 |
| | A' | 105 | 135 | 5Y 3/2 | MO, SB, ME | 10 | F, F | 12.0 | 7.8 | 6.2 | 105 | 22.2 | 1.2 | 1.2 |
| | Bw | 135 | 170 | 2.5Y 3/2 | MO, SB, ME | 4 | F, F | 7.1 | 7.8 | 5.9 | 86 | 19.3 | 1.3 | 1.3 |
| | ˆAum | 0 | 20 | 2.5Y 5/3 | MA | 10 | absent | 19.4 | 8.0 | 7.0 | 239 | 23.1 | 5.6 | 5.6 |
| | Aum | 20 | 65 | 2.5Y 5/3 | MA | 8 | absent | 14.5 | 6.7 | 5.9 | 418 | 22.2 | 0.0 | 0.0 |
| | Bwm | 65 | 80 | 2.5Y 5/4 | MA | 6 | absent | 7.5 | 7.2 | 5.9 | 214 | 20.3 | 0.0 | 0.0 |
| P2 | Bw1 | 80 | 100 | 2.5Y 5/4 | MO, SB, ME | 4 | absent | 4.8 | 7.4 | 6.0 | 127 | 14.2 | 0.0 | 0.0 |
| | Bw2 | 100 | 150 | 2.5Y 4/3 | MO, SB, ME | 6 | absent | 5.1 | 7.6 | 6.2 | 71 | 16.8 | 0.0 | 0.0 |
| | Bw3 | 150 | 175 | 2.5Y 4/3 | MO, SB, ME | 3 | absent | 4.8 | 7.8 | 6.2 | 61 | 14.1 | 0.0 | 0.0 |
| | Ab | 175 | 185+ | 10YR 4/3 | MO, SB, ME | 1 | absent | 17.9 | 7.7 | 6.4 | 95 | 50.2 | 0.0 | 0.0 |
| | ˆAum1 | 0 | 10 | 2.5Y 4/2 | MA | 12 | VF, F | 21.7 | 7.8 | 6.9 | 255 | 24.7 | 5.9 | 5.9 |
| | ˆAum2 | 10 | 25 | 2.5Y 4/2 | MA | 8 | VF, F | 23.7 | 7.1 | 6.3 | 277 | 26.2 | 1.8 | 1.8 |
| P3 | A | 25 | 45/75 | 2.5Y 4/4 | MO, SB, ME | 6 | VF, F | 15.6 | 7.1 | 5.8 | 116 | 24.7 | 0.0 | 0.0 |
| | Bw1 | 45/75 | 105 | 2.5Y 4/4 | MO, SB, ME | 2 | VF, F | 8.6 | 7.5 | 5.9 | 60 | 22.2 | 0.0 | 0.0 |
| | Bw2 | 105 | 160+ | 2.5Y 4/4 | MO, SB, ME | 1 | VF, F | 8.4 | 7.6 | 6.1 | 55 | 22.9 | 0.0 | 0.0 |

**Table 3.** *Cont.*

| | | Depth (cm) | | Soil Color | Structure | Coarse Fragments | Roots | OC | pH | | | CEC | TC | EC |
|---|---|---|---|---|---|---|---|---|---|---|---|---|---|---|
| Profile | Soil Horizon | Up | Down | | | % | | g/kg | H$_2$O | KCl | NaF | cmol(+)/kg | % | µS cm$^{-1}$ |
| | ^Aum1 | 0 | 25 | 2.5Y 4/2 | MA | 8 | F, C | 14.7 | 8.4 | 7.0 | 10.0 | 22.3 | 4.0 | 102 |
| | ^Aum2 | 25 | 50 | 2.5Y 4/3 | MA | 6 | F,C | 20.2 | 8.1 | 7.1 | 10.1 | 21.8 | 6.3 | 75 |
| | ^Cu | 50 | 70 | 2.5Y 5/2 | MA | 70 | VF, C | 13.6 | 8.3 | 7.5 | 10.4 | 15.0 | 47.8 | ND |
| P4 | 2Bwm | 70 | 110 | 10YR 4/4 | MA | 4 | VF, F | 17.4 | 7.8 | 6.3 | 10.0 | 25.9 | 1.0 | 72 |
| | 2Bw1 | 110 | 140 | 10YR 4/4 | MO, SB, ME | absent | VF, F | 17.5 | 7.5 | 6.2 | 9.9 | 37.7 | 0.0 | 105 |
| | 2Bw2/Ab | 140 | 160 | 10YR 3/3 | MO, SB, ME | absent | VF, F | 21.5 | 7.5 | 6.2 | 9.9 | 49.8 | 0.0 | 101 |
| | 2Bwb | 160 | 180+ | 10YR 4/4 | MO, SB, ME | absent | VF, F | 4.4 | 7.8 | 6.1 | 9.9 | 31.2 | 0.0 | 76 |
| | ^Aum | 0 | 15 | 2.5Y 4/2 | MA | 3 | F, F | 16.9 | 8.1 | 7.1 | 10.0 | 22.1 | 10.7 | 328 |
| | Aum1 | 15 | 45 | 2.5Y 4/2 | MA | 2 | F, F | 14.7 | 7.4 | 6.1 | 9.9 | 23.8 | 0.0 | 135 |
| | Aum2/Cm | 45 | 63 | 2.5Y 4/2 | MA | 3 | absent | 7.4 | 7.6 | 5.9 | 10.0 | 22.2 | 0.0 | 75 |
| | Cm | 63 | 70 | 5Y 6/2 | MA | absent | absent | 3.8 | ND | ND | ND | 10.1 | ND. | 45 |
| P5 | Am | 70 | 80 | 10YR 4/2 | MA | absent | F, F | 8.8 | 7.5 | 5.9 | 10.0 | 20.5 | 0.0 | 53 |
| | AB | 80 | 100 | 10YR 4/2 | MO, SB, ME | absent | absent | 11.1 | 7.4 | 5.9 | 9.9 | 22.4 | 0.0 | 60 |
| | AB/Bw | 100 | 125 | 10YR 4/3 | MO, SB, ME | absent | absent | 8.7 | 7.4 | 6.0 | 9.9 | 19.8 | 0.0 | 58 |
| | Ab | 125 | 170 | 10YR 3/3 | MO, SB, ME | absent | absent | 22.1 | 7.5 | 6.1 | 9.9 | 45.7 | 0.0 | 73 |
| | Bwb | 170 | 200+ | 10YR 4/4 | MO, SB, ME | absent | absent | 3.6 | 7.8 | 6.0 | 9.9 | 31.9 | 0.0 | 70 |
| | ^Aum | 0 | 35 | 2.5Y 5/3 | MA | 3 | VF, F | 18.9 | 7.8 | 6.9 | 10.1 | 24.4 | 9.8 | 281 |
| | Aum | 35 | 60/70 | 2.5Y 4/3 | MA | 12 | F, F | 13.3 | 7.6 | 6.8 | 10.0 | 23.0 | 6.3 | 233 |
| P6 | Bw | 60/70 | 95 | 2.5Y 4/3 | MO, SB, ME | 2 | absent | 10.0 | 7.6 | 6.1 | 9.9 | 23.5 | 0.0 | 79 |
| | 2Ab1 | 95 | 120 | 10YR 3/2 | MO, SB, ME | absent | absent | 23.4 | 7.1 | 6.0 | 9.8 | 49.9 | 0.0 | 155 |
| | 2Ab2 | 120 | 150 | 10YR 3/2 | MO, SB, ME | absent | absent | 23.3 | 7.0 | 5.9 | 9.9 | 53.0 | 0.0 | 169 |
| | 2Bwb | 150 | 160+ | 10YR 4/6 | MO, SB, ME | 1 | absent | 3.9 | 7.3 | 5.8 | 9.9 | 29.3 | 0.0 | 148 |
| | Ap | 0 | 10/15 | 10YR 4/3 | MO, SB, ME | absent | F, C | 17.8 | 7.5 | 6.0 | 9.8 | 45.3 | 0.0 | 150 |
| P7 | A | 10/15 | 65 | 10YR 3/2 | MO, SB, ME | absent | F, F | 21.4 | 7.3 | 5.9 | 9.9 | 51.0 | 0.0 | 171 |
| | Bw | 65 | 85 | 10YR 4/6 | MO, SB, ME | 3 | VF, F | 8.5 | 7.1 | 5.9 | 9.9 | 27.8 | 0.0 | 120 |
| | BC | 85 | 100+ | 2.5Y 4/4 | WE, SB, ME | 8 | absent | 0.8 | 7.2 | 6 | 9.7 | 15.3 | 0.0 | 135 |

Legend: color: 2.5Y 4/2 = dark grayish brown; 2.5Y 4/3 and 4/4 = olive brown; 2.5Y 5/2 = grayish brown; 2.5Y 5/3 = light olive brown; 10YR 3/2 = very dark grayish brown; 10YR 3/3 = dark brown; 10YR 4/3 = brown; 10YR 4/4 and 4/6 = dark yellowish brown; 5Y 3/2 = dark olive gray; 5Y 4/2 = olive gray; 5Y 4/4, 5/3 and 5/4 = olive; structure: degree of development: WE = weak, MO = moderate; types: MA = massive, SG = single grain, AB = angular blocky, SB = subangular blocky, PR = prismatic, GR = granular, PL = platy, CR = crumbly; size classes (mm) for blocky/crumbly/lumpy/cloddys: VF = very fine/thin < 5, FI = fine/thin 5–10, ME = medium 10–20, CO = coarse/thick 20–50, VC = very coarse/thick > 50; roots: diameter (mm): VF = very fine < 0.5, F = fine 0.5–2, M = medium 2–5, C = coarse > 5; abundance (number): N = none 0, V = very few = 1–20, F = few 20–50, C = common 50–20, M = many > 200; OC = organic carbon, CEC = cation exchange capacity; TC = total carbonates; EC = electrical conductivity.

**Table 4.** (**A**,**B**) Data of selected PTEs content (mg/kg), measured using the pXRF technique in situ, on soil profiles. Underlined: reported data exceeding CSC for residential/recreational land use (CSC1). Italics: values exceeding CSC for agricultural land use (CSC2). Bold: values exceeding CSC for commercial/industrial land use (CSC3). For As, recalculated values based on Caporale et al., 2018 are reported in brackets. ND = not determined. The caret symbol (^) is used as a prefix to indicate horizons and layers formed in human-transported material.

**(A)**

| Profile | Soil Horizon | Horizon Depth | | Depth of Measurement | As | Cd | Cr | Cu | Pb | Sn | Zn |
|---|---|---|---|---|---|---|---|---|---|---|---|
| | | Up | Down | | | | | mg/kg | | | |
| | ^Au | 0 | 25 | 5 | **74 (47) ± 10** | **24** | 35 ± 3 | 111 ± 4 | **3326 ± 22** | 59 ± 8 | 107 ± 3 |
| | ^Aum1 | 25 | 45 | 30 | **73 (46) ± 18** | **46** | 43 ± 4 | 257 ± 7 | **6791 ± 50** | 124 ± 11 | 145 ± 5 |
| | ^Aum2 | 45 | 60 | 50 | **63 (42) ± 7** | **70** | 22 ± 3 | 42 ± 3 | **1748 ± 13** | 33 ± 8 | 70 ± 3 |
| | ^Aum3 | 60 | 85 | 60 | ND | **34** | 22 ± 3 | 46 ± 3 | **3839 ± 25** | 71 ± 8 | 85 ± 3 |
| P1 | | | | 80 | *34 (28) ± 9* | **25** | 31 ± 3 | *176 ± 5* | **3154 ± 20** | 41 ± 8 | <u>172 ± 4</u> |
| | A | 85 | 100 | | | | | | | | |
| | A′ | 105 | 135 | 110 | <u>24 (24)</u> ± 3 | ND | 24 ± 4 | 73 ± 4 | *294 ± 4* | ND | 91 ± 3 |
| | | | | 150 | <u>24 (24)</u> ± 3 | ND | 24 ± 3 | 68 ± 4 | *290 ± 4* | ND | 74 ± 3 |
| | Bw | 135 | 170+ | 170 | 16 <u>(20)</u> ± 2 | ND | 32 ± 4 | 27 ± 3 | 60 ± 3 | ND | 64 ± 3 |
| | ^Aum1 | 0 | 20 | 5 | ND | **57** | 54 ± 4 | *210 ± 6* | **6242 ± 39** | 74 ± 9 | <u>183 ± 5</u> |
| | | | | 20 | **141 (77) ± 13** | **47** | 32 ± 4 | *193 ± 6* | **4749 ± 32** | 70 ± 9 | 150 ± 4 |
| P2 | Aum | 20 | 65 | 40 | <u>22 (23)</u> ± 4 | ND | 39 ± 4 | 44 ± 4 | *259 ± 5* | ND | 83 ± 3 |
| | Bwm | 65 | 80 | 65 | 14 <u>(19)</u> ± 2 | ND | 23 ± 4 | 31 ± 3 | 68 ± 3 | ND | 66 ± 3 |
| | Bw1 | 80 | 100 | 80 | 16 (20) ± 2 | ND | 26 ± 3 | 33 ± 3 | 51 ± 2 | ND | 67 ± 3 |
| | Bw2 | 100 | 150 | 110 | 13 (19) ± 2 | ND | 25 ± 3 | 28 ± 3 | 48 ± 2 | ND | 63 ± 3 |
| | Ab | 175 | 185+ | 180 | 19 <u>(22)</u> ± 1 | ND | 21 ± 3 | 11 ± 2 | 32 ± 2 | ND | 40 ± 2 |
| | ^Aum1 | 0 | 10 | 3 | ND | **106** | 67 ± 4 | *330 ± 7* | **12,198 ± 75** | 141 ± 9 | <u>215 ± 5</u> |
| | ^Aum2 | 10 | 25 | 12 | ND | **84** | 64 ± 4 | *323 ± 7* | **11,071 ± 67** | 140 ± 9 | <u>290 ± 6</u> |
| P3 | A | 25 | 45/75 | 30 | *41 (32) ± 5* | ND | 38 ± 3 | 102 ± 4 | *709 ± 7* | ND | 112 ± 3 |
| | Bw1 | 45/75 | 105 | 60 | 17 <u>(21)</u> ± 2 | ND | 33 ± 3 | 42 ± 3 | 68 ± 2 | ND | 75 ± 3 |
| | | | | 110 | 14 (19) ± 2 | ND | 41 ± 4 | *185 ± 5* | 57 ± 2 | ND | 69 ± 3 |
| | Bw2 | 105 | 160+ | 150 | 16 (20) ± 2 | ND | 26 ± 4 | 26 ± 3 | 60 ± 3 | ND | 76 ± 3 |
| | | | | 160 | <u>23 (23)</u> ± 2 | ND | 32 ± 3 | 33 ± 3 | *107 ± 3* | ND | 86 ± 3 |

**Table 4.** *Cont.*

| | | | | | **(B)** | | | | | | |
|---|---|---|---|---|---|---|---|---|---|---|---|
| Profile | Soil Horizon | Horizon Depth Up | Horizon Depth Down | Depth of Measurement | As | Cd | Cr | Cu mg/kg | Pb | Sn | Zn |
| | | | | 0 | *37 (30)* ± 10 | **32** | 40 ± 3 | 95 ± 4 | **3186 ± 21** | 64 ± 8 | 115 ± 3 |
| | ^Aum1 | 0 | 25 | 10 | *36 (29)* ± 6 | **18** | 20 ± 4 | 27 ± 3 | **1002 ± 9** | 29 ± 8 | 55 ± 3 |
| | | | | 20 | ND | **22** | ND | ND | **1117 ± 19** | 39 ± 10 | 63 ± 9 |
| P4 | ^Aum2 | 25 | 50/55 | 35 | ND | **23** | 29 ± 4 | 148 ± 5 | **3857 ± 28** | 93 ± 10 | 135 ± 4 |
| | ^Cu | 50/55 | 75/80 | 50 | 21 (23) ± 3 | ND | 15 ± 3 | 51 ± 4 | *117 ± 4* | ND | 21 ± 2 |
| | 2Bwm | 75/80 | 110 | 75 | 17 (21) ± 2 | ND | 33 ± 3 | 47 ± 3 | 63 ± 2 | ND | 67 ± 3 |
| | 2Bw2/Ab | 140 | 160 | 155 | ND | ND | ND | ND | 42 ± 4 | ND | 29 ± 9 |
| | ^Aum | 0 | 15 | 0 | **183 (96) ± 16** | 43 | 34 ± 3 | 145 ± 5 | **7350 ± 45** | 80 ± 9 | <u>220 ± 5</u> |
| | | | | 10 | **143 (78) ± 15** | 33 | 34 ± 4 | *146 ± 5* | **6058 ± 39** | 85 ± 9 | <u>204 ± 5</u> |
| | Aum1 | 15 | 40 | 20 | 15 (20) ± 2 | ND | 34 ± 3 | 56 ± 4 | *142 ± 3* | ND | 84 ± 3 |
| | | | | 35 | 18 (21) ± 2 | ND | 24 ± 4 | 42 ± 3 | 79 ± 3 | ND | 78 ± 3 |
| | Aum2/Cm | 45 | 63 | 55 | ND | ND | | ND | 40 ± 5 | ND | 75 ± 11 |
| P5 | Cm | 63 | 70 | 65 | ND | ND | | ND | 39 ± 5 | ND | 74 ± 12 |
| | Am | 70 | 80 | 70 | 18 (21) ± 2 | ND | 33 ± 4 | 54 ± 4 | 38 ± 2 | ND | 55 ± 3 |
| | AB | 80 | 100 | 100 | 22 (23) ± 2 | ND | 22 ± 3 | 15 ± 3 | 32 ± 2 | ND | 45 ± 2 |
| | AB/Bw | 100 | 125 | | | | ± | | | | |
| | Ab | 125 | 185 | 135 | 18 (21) ± 1 | ND | 24 ± 3 | 13 ± 2 | 31 ± 2 | ND | 35 ± 1.9 |
| | Bwb | 185 | 200+ | 185 | 14 (19) ± 2 | ND | 23 ± 4 | ND | 40 ± 2 | ND | 45 ± 2 |
| | ^Aum | 0 | 35 | 5 | **144 (78) ± 19** | 63 | 33 ± 3 | *246 ± 6* | **9695 ± 60** | 141 ± 9 | <u>185 ± 5</u> |
| | | | | 10 | ND | 63 | ND | ND | **10,508 ± 135** | 148 ± 11 | 147 ± 11 |
| P6 | | | | 20 | **99 (58) ± 10** | 19 | 34 ± 3 | 91 ± 4 | **3392 ± 22** | 43 ± 8 | 132 ± 4 |
| | Aum | 35 | 60/70 | 40 | 13 (19) ± 3 | ND | ND ± | 39 ± 4 | *123 ± 4* | ND | 71 ± 3 |
| | Bw | 60/70 | 95 | 70 | 21 (23) ± 2 | ND | 33 ± 3 | 14 ± 3 | 40 ± 2 | ND | 61 ± 3 |
| P7 | A1 | 0 | 10 | 5 | 10 (18) ± 1 | ND | 30 ± 4 | 89 ± 4 | *869 ± 6* | ND | 97 ± 3 |
| | A2 | 10 | 65 | 20 | ND | ND | 25 ± 3 | 15 ± 3 | *256 ± 4* | ND | 30 ± 3 |

Legend of PTEs limits of each CSC. CSC1: As: 20 mg/kg; Cd: 2 mg/kg; Cr: 150 mg/kg; Cu: 120 mg/kg; Pb: 100 mg/kg; Sn: 1 mg/kg; Zn: 150 mg/kg; CSC2: As: 30 mg/kg; Cd: 5 mg/kg; Cr: 150 mg/kg; Cu: 200 mg/kg; Pb: 100 mg/kg; Zn: 300 mg/kg; CSC3: As: 50 mg/kg; Cd: 15 mg/kg Cr: 800 mg/kg; Cu: 600 mg/kg; Pb: 1000 mg/kg; Sn: 350 mg/kg; Zn: 1500 mg/kg.

### 3.4. PXRF (Portable X-ray Fluorescence): Spatial Survey

Preliminary analyses were performed on fine blackish and greyish fine fractions (less than 2 mm) obtained by sieving the materials collected from the waste hills in place in site 1 before the rearrangement works. Strong contamination, exceeding CSC3, was found for Pb and Cd (Pb from 1860 to 15,121 mg/kg and Cd from 24 to 150 mg/kg), while for Sn (Sn from 57 to 273 mg/kg), values did not exceed CSC3 values.

Furthermore, among the pXRF measurements performed on the samples collected on the regular grid at two depths (Table 5), only the mean values of As, Cd, Pb and Sb exceeded the CSC3 at both depths. Cu showed hot spots (3382 mg/kg), mainly in the topsoil, as well as Sn (1682 mg/kg). Bivariate Pearson analysis applied to pXRF measurements at 0–10 cm depth showed significantly positive correlation coefficients (two-tails, ** at 0.01 level and * at 0.05 level) between Pb and Sb (0.950 **), Cd (0.840 **), Ni (0.742 **), As (0.750 **), Sn (0.715 **), Fe (0.620 **), Zn (0.546 **) and Cu (0.290 **) (Table S1A). At the next depth (10–40 cm), some correlation coefficients notably increased, such as that between Pb and Sn (0.772 **), Fe (0.753 **), Zn (0.671 **) and Cu (0.533 **) (Table S1B). On the contrary, negative correlations were found at 0–10 cm between Pb and Mn (−0.709 **), Ti (−0.659 **), K (−0.637 **), Zr (−0.535 **), V (−0.420 *), Sr (−0.370 *), Rb (−0.324 *) and Th (−0.301 *). The positive correlations found between Pb and many PTEs suggested a common source of contamination. Indeed, different types of batteries were stocked as waste in site 1 of the studied area, including Pb (which contained the conductive support grid made by Pb-Sb alloys), Ni-Cd and Ni-Zn batteries. Concerning As: considering the high levels reached (mostly in site 1), anthropogenic sources, including residues of industrial processes of battery assemblage, must be taken into account [73]. Indeed, even though natural origin has been reported for As in these environments [60,74], levels in the study site were excessively high and attributable mainly to pollutant sources. Regarding the negative correlations found between Pb and Ti, Th, Zr, and Nb, these elements are known to naturally enrich volcanic soils of the Phlegrean Fields, Mt. Roccamonfina volcano and Nola-Pomigliano areas (north of Mt. Somma-Vesuvius volcano) [60]. Additionally, K (and its vicariant Rb) [75] and the Zr/Y, Nb/Y, Nb/Zr and Th/Ta ratios are used for geochemical signatures

of specific volcanic areas in central Italy [76]. Therefore, the negative correlations between Pb and Ti, Th, Zr and Nb supported the hypothesis of anthropogenic and natural origins of these elements, respectively.

Among the elements exceeding the CSC3, Cd, Pb and As were selected to be mapped (Figure 7) for validation of the PSS method used for HZs identification. In all the maps, values were divided into groups following the CSC provided by the Italian regulation [36,37]. Hot spots of Cd contamination (in the range 100–200 mg/kg) were principally concentrated in site 1, in both HZ1 and HZ2, and a similar trend was observed for site 2 where some hot spots of Cd were found in proximity to the removed waste hills (in HZ3 and HZ4a) (Figure 7A). In correspondence with HZ4b, considerably lower concentrations of Cd were detected, with areas reaching levels below CSC2 and CSC1. However, contamination levels generally decreased with depth, although strong Cd concentrations persisted in HZ1. Meanwhile, the less contaminated areas in HZ4b grew larger with depth (Figure 7B). Concerning Pb distribution: all of site 1 was widely and strongly contaminated, with measured values exceeding the CSC3 by ten to thirty times (Figure 7C). High consistency was found between Pb and Cd distribution for both sites 1 and 2. With depth (Figure 7D), weak decreases of Pb contamination were observed, as well as an increased extension of the less contaminated area in HZ4b. The spatial distribution of As (Figure 7E) was similar; it was highly concentrated in site 1, with the highest concentrations in the southern part of HZ1, as well as in HZ3 and partially in HZ4a, while As concentrations below CSC1 and CSC2 were found mainly in HZ4b. In both site 1 and site 2, decreased As concentrations were detected with depth (Figure 7F).

**Table 5.** Descriptive statistics on the content of 20 elements measured by pXRF (portable X-ray fluorescence) spectrometer at two soil depths (0–10 and 10–40 cm). Underlined: reported data exceeding CSC for residential/recreational land use (CSC1). Italics: values exceeding CSC for agricultural land use (CSC2). Bold: values exceeding CSC for commercial/industrial land use (CSC3). For As, recalculated values based on Caporale et al., 2018 were reported.

| | As | Cd | Cr | Cu | Ni mg/kg | Pb | Sn | Zn | V | Mn | Nb | Rb | Sr | Sb mg/kg | Th | Ti | Zr | Ca | Fe g/kg | K |
|---|---|---|---|---|---|---|---|---|---|---|---|---|---|---|---|---|---|---|---|---|
| **0–10 cm** | | | | | | | | | | | | | | | | | | | | |
| N. cases | 91 | 91 | 120 | 120 | 118 | 120 | 107 | 120 | 120 | 120 | 119 | 120 | 120 | 101 | 120 | 120 | 120 | 120 | 120 | 120 |
| Mean | **69** | **74** | 60 | *335* | 35 | **9456** | *178* | *226* | *115* | 1055 | 59 | 252 | 533 | **229** | 39 | 3126 | 386 | 32.8 | 38.0 | 29.2 |
| St. Dev. | 45 | 41 | 12 | 448 | 21 | 7888 | 195 | 118 | 9 | 111 | 6 | 25 | 45 | 148 | 6 | 338 | 36 | 19.7 | 7.7 | 4.8 |
| Min | 21 | **16** | 38 | 55 | 15 | *135* | 27 | 91 | *93* | 589 | 28 | 96 | 328 | *30* | 19 | 1991 | 174 | 17.9 | 27.0 | 15.9 |
| Max | **230** | 185 | 125 | **3382** | *210* | 45,310 | **1682** | *969* | *150* | 1402 | 77 | 368 | 670 | **854** | 54 | 4345 | 453 | 224.7 | 100.7 | 44.9 |
| Coef. Var. % | 64.7 | 56.0 | 19.7 | 133.7 | 59.7 | 83.4 | 109.4 | 52.4 | 8.2 | 10.5 | 10.2 | 9.9 | 8.4 | 64.6 | 15.6 | 10.8 | 9.3 | 60.1 | 20.1 | 16.4 |
| Skewness | 1.3 | 0.5 | 2.7 | 4.7 | 5.5 | 1.2 | 5.3 | 3.7 | 0.5 | −0.1 | −1.7 | −1.8 | −0.4 | 1.3 | −0.5 | 0.1 | −2.5 | 8.1 | 5.2 | 0.3 |
| Kurtosis | 1.6 | −0.6 | 13.6 | 25.7 | 41.7 | 2.6 | 36.0 | 19.7 | 0.5 | 1.9 | 9.8 | 17.2 | 3.8 | 3.2 | 0.9 | 0.7 | 12.5 | 77.0 | 38.7 | −0.3 |
| **10–40 cm** | | | | | | | | | | | | | | | | | | | | |
| N. cases | 98 | 59 | 98 | 98 | 94 | 98 | 76 | 98 | 98 | 98 | 98 | 98 | 98 | 73 | 98 | 98 | 98 | 98 | 98 | 98 |
| Mean | **56** | **57** | 56 | *217* | 29 | **5781** | *132* | *184* | *118* | 1111 | 59 | 257 | 550 | **166** | 40 | 3269 | 398 | 31.2 | 35.8 | 31.3 |
| St. Dev. | 40 | 39 | 9 | 184 | 11 | 6241 | 122 | 70 | 9 | 116 | 5 | 20 | 43 | 130 | 6 | 332 | 33 | 10.9 | 3.6 | 4.3 |
| Min | 18 | 16 | 38 | 42 | 15 | 100 | 25 | 94 | 86 | 813 | 41 | 173 | 409 | **32** | 19 | 2234 | 290 | 19.5 | 27.0 | 19.9 |
| Max | **211** | **200** | 82 | **859** | 62 | **30,819** | 772 | *399* | *140* | 1301 | 74 | 301 | 668 | **578** | 56 | 3852 | 492 | 112.2 | 49.1 | 39.6 |
| Coef. Var. % | 70.8 | 68.8 | 16.7 | 84.9 | 35.8 | 108.0 | 91.8 | 38.1 | 7.8 | 10.5 | 8.7 | 7.7 | 7.9 | 78.2 | 13.8 | 10.2 | 8.2 | 35.0 | 10.1 | 13.7 |
| Skewness | 1.9 | 1.5 | 0.9 | 1.9 | 1.2 | 1.5 | 2.5 | 0.9 | −0.7 | −0.5 | −0.9 | −1.2 | −0.2 | 1.2 | −0.3 | −1.0 | −0.8 | 4.5 | 1.2 | −0.9 |
| Kurtosis | 3.8 | 2.2 | 0.9 | 3.7 | 1.2 | 2.1 | 9.5 | 0.9 | 0.0 | −0.5 | 2.5 | 4.2 | 0.5 | 0.6 | 2.1 | 0.1 | 2.2 | 30.8 | 2.6 | −0.2 |

Legend of PTEs limits of CSC1, CSC2, CSC3. CSC1: As: 20 mg/kg; Cd: 2 mg/kg; Cr: 150 mg/kg; Cu: 120 mg/kg; Ni: 120 mg/kg; Pb: 100 mg/kg; Sb: 10 mg/kg; Sn: 1 mg/kg; V: 90 mg/kg; Zn: 150 mg/kg; CSC2: As: 30 mg/kg; Cd: 5 mg/kg; Cr: 150 mg/kg; Cu: 200 mg/kg; Ni: 120 mg/kg; Pb: 100 mg/kg; Sb: 10 mg/kg; V:90 mg/kg; Zn: 300 mg/; CSC3: As: 50 mg/kg; Cd: 15 mg/kg Cr: 800 mg/kg; Cu: 600 mg/kg; Ni: 500 mg/kg; Pb: 1000 mg/kg; Sb: 30 mg/kg; Sn: 350 mg/kg; V: 250 mg/kg; Zn: 1500 mg/kg.

Bivariate Pearson correlation analysis was applied to penetrometry (CI), γ-rays dose rate and pXRF measurements (Table S1A,B). CI values in the first 10 cm showed significant correlation coefficients (at 0.05 level, two-tails), with only Pb (0.293 *), Zn (0.293 *) and Sn (0.292 *) measured in the 0–10 cm of depth, suggesting that the more hardened the soil

surface (i.e., higher CI values), the more anthropized/contaminated the soil itself. However, generally stronger correlation coefficients were found between γ-rays dose rate and the content levels of several PTEs measured in the depth of 0–10 cm (Table S1A), including Pb (−0.372 **), Cu (−0.228 **), Zn (−0.265 *) and As (−0.308 *). In the next depth, 10–45 cm, higher correlations were found between γ-rays and some PTEs, such as Zn (−0.374 **) and Cu (−0.315 **), while a new correlation was found with Ni (−0.242 *) (Table S1B). The opposite trends between γ-rays dose rate and Pb, Zn, Cu, As and Ni were clearly due to the allochthonous source of these elements, mainly contained in anthropogenic waste materials, while γ-rays emissions were higher for autochthonous volcanic materials and elements. Indeed, positive correlation coefficients were identified at depths of 10–45 cm between γ-rays dose rate and V (0.511 **), Sr (0.491 **), Ti (0.451 **), Mn (0.436 **), K (0.390 **), Nb (0.319 **), Zr (0.317 **) and Rb (0.283 *), while correlation with Th (0.261 *) was found only in 0–10 cm of depth. On the whole, these results were consistent with the generally high values of γ-rays dose rates emitted by volcanic materials, which are naturally enriched with minerals, bearing K, Ti, Mn, Sr, Zr and Th. The significant correlations found between γ-rays dose rate and anthropogenic elements were due to their concentration in the upper 10 to 40 cm of soil. Indeed, according to previous works [77–79], 90% of γ-rays radiation is emitted from the first 20 to 30 cm of rock and from the first 30 to 50 cm of soil depth. Therefore, the contribution from deeper rocks and soils was negligible.

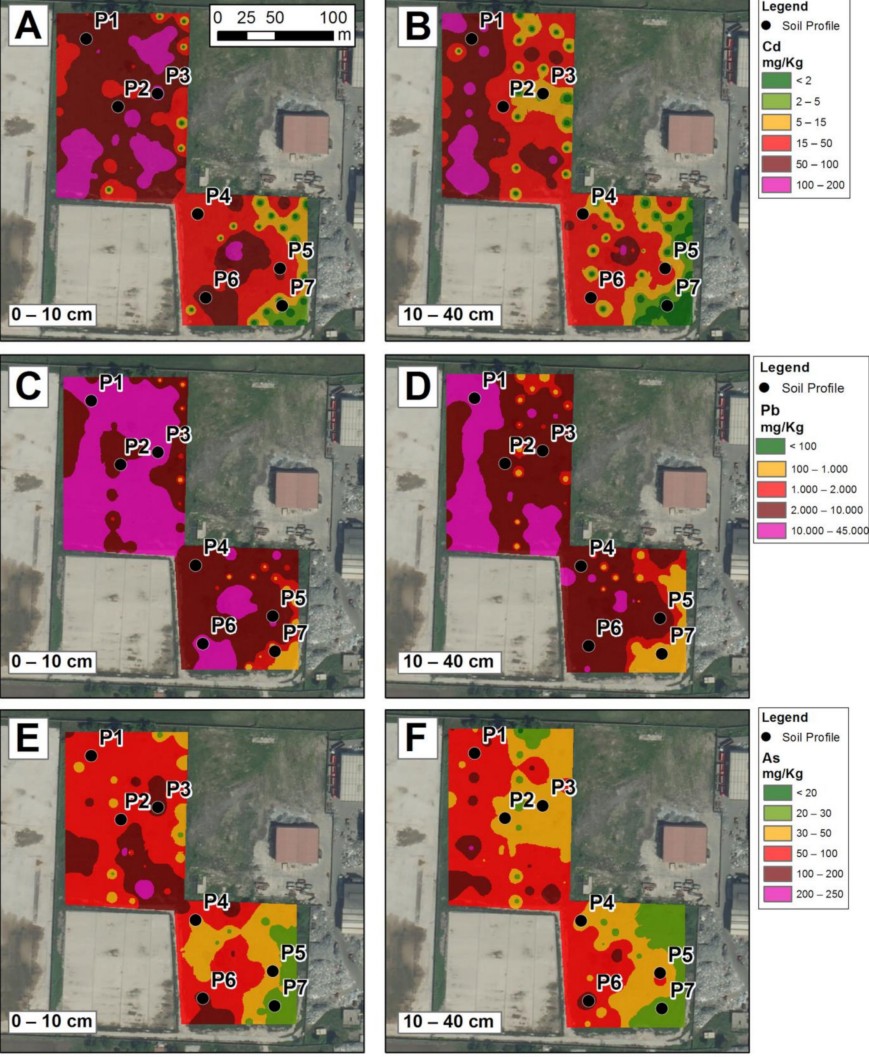

**Figure 7.** Maps of PTE distribution, measured by PXRF. (**A**) Cd at 0–15 cm; (**B**) Cd at 15–40 cm; (**C**) Pb at 0–15 cm; (**D**) Pb at 15–40 cm; (**E**) As at 0–15 cm; (**F**) As at 15–40 cm.

## 4. Conclusions

In the framework of characterization and remediation strategies of potentially contaminated sites, this work tested the effectiveness of different PSS methods and combined them with pedological investigation in order to facilitate spatial variability assessment of the physical and chemical degradation phenomena affecting soil in an industrial site in southern Italy. The use of the EMI and ARP techniques—provided by the Italian regulatory body for potentially polluted sites with unknown spatial distribution of contaminants—in combination with γ-rays and ultrasound penetrometry surveys proved very effective for the identification of HZs. This was verified and validated through the use of pXRF for elemental analyses. Moreover, the detailed pedological characterization enabled us to identify high variability in the in-depth soil stratigraphy, as well as extensive soil physical degradation and PTE (mainly Pb, As, Cd) contamination from the soil surface down to variable soil depths. The validation analysis, obtained using pXRF spectrometry, and the high consistency between γ-rays and pXRF maps, demonstrated that the most successfully applied technique in this study was γ-rays spectrometry. This supported the use of natural and artificial radioactivity-based methodologies as proxy for the fast, noninvasive identification of soil degradation and pollution spatial variability.

Therefore, a multi-sensor approach must be increasingly considered a powerful tool for the fast identification of HZs in degraded environments. It offers the tangible opportunity to address targeted prevention and remediation strategies. This study confirmed and highlighted the fact that site-specific investigation strategies must be tuned every time environmental studies are faced. A priori, neither the best performing or the most suitable proximal sensors exist; only the use of combined techniques based on different physical principles associated with pedological surveys will enable researchers to describe, as closely as possible, the real extent and variability, both in space and depth, of soil degradation phenomena.

**Supplementary Materials:** The following supporting information can be downloaded at: https://www.mdpi.com/article/10.3390/app12083993/s1. Table S1. (**A**) Pearson correlation coefficients between elements measured on the regular grid at a depth of 0–10 cm by pXRF (Fe, K, Pb, Ti, Mn, Sr, Sb, Cu, Zn, Zr, Sn, Rb, As, Cd, V, Cr, Ni, Nb, Th), and γ-rays (dose rate) and cone index data (CI); (**B**) Pearson correlation coefficients between elements measured on the regular grid at a depth of 10–45 cm by pXRF (Fe, K, Pb, Ti, Mn, Sr, Sb, Cu, Zn, Zr, Sn, Rb, As, Cd, V, Cr, Ni, Nb, Th), and γ-rays (dose rate) and cone index data (CI).

**Author Contributions:** Conceptualization, S.V., F.T., G.L. and P.M.; Methodology, S.V., F.T. and P.M.; Software, A.A. and F.A.M.; Validation, S.V., F.T., G.L. and P.M.; Formal Analysis, A.A., R.D.M., P.M. and S.V.; Investigation, A.A., S.V., R.D.M. and F.T.; Resources, F.T.; Data Curation, S.V., A.A. and F.A.M.; Writing—Original Draft Preparation, S.V. and F.T.; Writing—Review & Editing, S.V. and F.T.; Visualization, A.A., F.A.M. and S.V.; Supervision, S.V. and F.T.; Project Administration, F.T.; Funding Acquisition, F.T. All authors have read and agreed to the published version of the manuscript.

**Funding:** This research activity was funded by the agreement between ECOBAT S.p.a. and CIRAM—University of Naples Federico II, an Italian funding programme after-LIFE11/ENV/IT/275 EcoRemed project (2012–2017): Implementation of eco-compatible protocols for agricultural soil remediation in Litorale Domizio Agro Aversano NIPS.

**Conflicts of Interest:** The authors declare no conflict of interest.

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
