# Peer review of "Multi-Sensor Approach Combined with Pedological Investigations to Understand Site-Specific Variability of Soil Properties and Potentially Toxic Elements (PTEs) Content of an Industrial Contaminated Area"

_applsci, doi:10.3390/app12083993_

Round 1
Reviewer 1 Report
General comments
The manuscript is a very interesting example of the use of join geophysical and geochemical "in situ" methods for the complex characterization of contaminated soils. Such comprehensive characterization of site-specific variability physical and chemical properties is very helpful in developing the remediation strategy. The article is very well organized, precisely showing the advantages of each applied techniques and the obtained results are correlated with chemical data.
The only remarks concern the technical form of the presentation of the figures, which in most cases are too small to be legible (e.g. Figs. 1, 3c). The point numbers in Figures 5 and 6 are completely invisible and should be written in larger font.
Detailed remarks
Page 13, line 441. Instead of “There fore” should be “Therefore”
Page 13, line 418, page 14, line 457, page 15, line 492. Unjustified indentation in the text.
After these minor corrections, I consider the manuscript ready for publication.
Author Response
Dear reviewers,
on behalf of the authors of the article entitled “MULTI SENSORS APPROACH combined with pedological investigations to understand site-specific variability of soil properties and potentially toxic elements (PTEs) content of industrial contaminated soils” by Vingiani Simona, Agrillo Antonietta, De Mascellis Roberto, Langella Giuliano, Manna Piero, Mileti Florindo Antonio, Terribile Fabio
I wish to thank the reviewers for the careful and detailed revisions and suggestions, that strongly improved the quality of the manuscript, figures and tables.
All requests have been addressed and (as follows) answers are reported point by point, in black the reviewer request and in red the answer.
Associated with this revision note, the revised manuscript (tracked changes) has been uploaded.
Kind regards.
Simona Vingiani

Reviewer 2 Report
Dear Authors, you can found my comments in attached file.

Author Response
Dear reviewers,
on behalf of the authors of the article entitled “MULTI SENSORS APPROACH combined with pedological investigations to understand site-specific variability of soil properties and potentially toxic elements (PTEs) content of industrial contaminated soils” by Vingiani Simona, Agrillo Antonietta, De Mascellis Roberto, Langella Giuliano, Manna Piero, Mileti Florindo Antonio, Terribile Fabio
I wish to thank the reviewers for the careful and detailed revisions and suggestions, that strongly improved the quality of the manuscript, figures and tables.
All requests have been addressed and (as follows) answers are reported point by point, in black the reviewer request and in red the answer.
Associated with this revision note, the revised manuscript (tracked changes) has been uploaded. Just two last notes, also to answer to reviewer 2:
- all tables (word format) and figures will be uploaded in the requested format. In the figure boxes within the manuscript I had to past them as pictures to avoid that tables unformatted. I did of my best, but I have only limited editing/graphic abilities. I leave this work to MDPI graphic experts.
- I did not find correspondence between lines indicated by reviewers and lines of the submitted pdf manuscript. I also checked using the pdf received with results of plagiarism check, but no correspondence was found looking at this file too. However, I am confident that I’ve found all the right parts to amend, using some words indicated by the reviewers and interpretating their comments.
Kind regards.
Simona Vingiani

Round 2
Reviewer 2 Report
Dear Authors, I accepted paper for publication.